# Remote sensing estimation of sugar beet SPAD based on un-manned aerial vehicle multispectral imagery

**Weishi Gao**[1☯]**, WanYing Zeng**[2☯]**, Sizhong Li**[1]***, Liming Zhang**[1]**, Wei Wang**[3]**, Jikun Song**[4]**, Hao Wu**[3]

**1** Institute of Economic Crops, Xinjiang Academy of Agricultural Sciences, Urumqi, China, **2** College of Agronomy, Xinjiang Agricultural University, Urumqi, China, **3** Anyang Institute of Technology, AnYang, China, **4** Cotton Research Institute, Chinese Academy of Agricultural Sciences, AnYang, China

☯ These authors contributed equally to this work.

* xjnkylsz@163.com

**Data Availability Statement:** All relevant data are within the paper and its Supporting information files.

## Abstract

Accurate, non-destructive and cost-effective estimation of crop canopy Soil Plant Analysis De-velopment(SPAD) is crucial for precision agriculture and cultivation management. Unmanned aerial vehicle (UAV) platforms have shown tremendous potential in predicting crop canopy SPAD. This was because they can rapidly and accurately acquire remote sensing spectral data of the crop canopy in real-time. In this study, a UAV equipped with a five-channel multispectral camera (Blue, Green, Red, Red_edge, Nir) was used to acquire multispectral images of sugar beets. These images were then combined with five machine learning models, namely K-Nearest Neighbor, Lasso, Random Forest, RidgeCV and Support Vector Machine (SVM), as well as ground measurement data to predict the canopy SPAD of sugar beets. The results showed that under both normal irrigation and drought stress conditions, the SPAD values in the normal ir-rigation treatment were higher than those in the water-limited treatment. Multiple vegetation indices showed a significant correlation with SPAD, with the highest correlation coefficient reaching 0.60. Among the SPAD prediction models, different models showed high estimation accuracy under both normal irrigation and water-limited conditions. The SVM model demon-strated a good performance with a correlation coefficient (R2) of 0.635, root mean square error (Rmse) of 2.13, and relative error (Re) of 0.80% for the prediction and testing values under normal irrigation. Similarly, for the prediction and testing values under drought stress, the SVM model exhibited a correlation coeffi-cient (R2) of 0.609, root mean square error (Rmse) of 2.71, and rela-tive error (Re) of 0.10%. Overall, the SVM model showed good accuracy and stability in the pre-diction model, greatly facilitating high-throughput phenotyping research of sugar beet canopy SPAD.

## 1 Introduction

As one of the important sugar crops in China, sugar beet belonged to Amaranthaceae family and genus Beta [1], which was a biennial herb that undergoes nutritive growth in the first year

**Funding:** This research was funded by Key Research and development task special project of Xin-jiang(2022B02002); National Sugar Industry Technology System Project (CARS-170108); Henan Provincial Science and Technology Tackling Project(232102111132).

**Competing interests:** The authors have declared that no competing interests exist.

and reproductive growth in the second year after vernalisation. The rapid growth period of leaf tufts of sugar beet was from the beginning of the shedding of the root primordial cortex to the time when the daily growth of leaf tufts reaches the highest value, and the chlorophyll in this period provides abundant nutrients for sugar metabolism in the later underground part, and also photosynthesis reaches the highest level [2]. Chloro-phyll content is a highly signifi-cant indicator for photosynthesis and growth development in crops. It is a crucial pigment present in the chloroplasts of green plants and plays a major role in capturing energy during photosynthesis [3]. Its content was an important in-dicator of plant nutrient stress, nitrogen status, growth senescence and other stages, which is important for dynamic monitoring of veg-etation growth and rapid diagnosis of fertiliser application. SPAD can directly reflect the rela-tive content of chlorophyll in the leaves [4, 5]. Therefore, the study of SPAD content of sugar beet is of great significance for the growth of sugar beet. UAV remote sensing provides a new solution for the canopy traits of crops [6] and this study focuses with the SPAD prediction of sugar beet based on multispectral im-ages from UAV remote sensing [7].

In recent years, with the rapid development of multi-source sensors, big data and ar-tificial intelligence technologies, UAV remote sensing of agriculture has been developing rapidly in both breadth and depth [8, 9]. Compared with traditional orbital and satellite remote sensing, drones are more convenient and flexible, and because of the relatively economic cost, they have become the favoured object in the majority of research [10, 11]. In addition, the sensors carried by UAVs are also very diverse, such as infrared sensors, hy-perspectral sensors, multi-spectral sensors, etc [12, 13]. Due to the long processing time of multispectral and scientific research to achieve high-throughput phenotypic prediction, it has become a hotspot for large-scale crop canopy trait research. At present, many re-searchers use hyperspectral for the esti-mation of vegetation chlorophyll content has been a large number of studies. Hunt et al [14] constructed the vegetation index GNDVI through the multispectral images obtained by UAV, and inverted the leaf area index of wheat through the vegetation index. Wang et al [15] con-structed the SPAD inversion model of wheat through the selection of the contribution rate of the vegetation index by the use of UAV. Bendig et al. [16]and Yang et al. [17] predicted the biomass of crops by UAV and ob-tained better results.

The integration of low-altitude UAV remote sensing and ground-based measure-ments for modeling crop canopy traits has been a smart application of UAV remote sens-ing in precision agriculture [18, 19]. Machine learning, which is one of the earliest disci-plines in artificial intel-ligence focusing on numerical computation, allows for predicting unknown data based on established models and input parameters [20, 21]. Machine learning algorithms exhibit robust-ness and demonstrate high predictive capability not only in structured data with strong pheno-typic patterns but also in unstructured and non-linear domains [22, 23]. For instance, typical machine learning models such as Sup-port Vector Machines, Ridge Regression, Naive Bayes algorithm, and Decision Tree mod-els have shown significant advantages in agricultural remote sensing [24, 25].

Due to the powerful predictive ability of machine learning and the high throughput of UAV remote sensing in crop phenotyping, UAV remote sensing combined with ma-chine learning provides a good opportunity for research on crop canopy traits. Currently, there are few reports on the inversion of the canopy SPAD model based on sugar beets. In this study, we combined the UAV multispectral imagery with the extraction of sin-gle-band reflectance, and then constructed the vegetation index by combining with the vegetation index model, which provides a high correlation between the vegetation index and the SPAD to determine the parameters for inversion models and thus make predic-tions of the inversion model, with a view to achieving the intelligent monitoring of sugar beets.

## 2 Materials and methods

### 2.1 Experimental materials and design

The experiment was conducted in the sugar beet scale planting area of Manas Ex-perimental Station(86¡ã12'52.2"N, 44¡ã18'15.77"E) of Xinjiang Academy of Agricultural Sci-ences, Xin-jiang Changji Hui Autonomous Prefecture, Xinjiang, and 300 copies of domestic and foreign staple varieties were selected as the participating materials, which were planted in the morning of April 27, 2022. The Manas area has a mid-temperate continental arid-semi-arid climate, with severe cold winters, hot summers, dryness and little rainfall, abundant sunshine, high evapotranspiration, and low precipitation. In this study, two ir-rigation water treatments were set up: normal irrigation (NI) and drought stress (DS), 600 plots were selected for each irriga-tion treatment, and one variety was planted in each plot in a completely randomised block design with two replications, 3 row zones, 2 m row length and 20 cm row spacing, and drought stress was the water-control management for the reproductive period of the sugar beet. The field management was in accordance with the local conventional cultivation management, and the beets could grow normally and well in the field, and the fertilisation, drip irrigation, pest control and weeding were the same as the local field management.

### 2.2 Drone platforms and flight settings

The UAV used in this study is the DJI Genie 4 Multispectral Edition UAV P4M, which inte-grates one visible sensor channel and five multispectral sensor channels with five bands of multispectral channels centred on wavelengths of 450 nm (Blue), 560 nm (Green), 650 nm (Red), 730 nm (Red_edge), 840 nm (Nir). Each shot can obtain 6 images and each image has a resolution of more than 2 million pixels. At the same time, the UAV is equipped with Time-Sync time synchronisation system, which can obtain centimetre-level positioning accuracy. In addition, the integrated light intensity sensor on the top of the P4M can capture solar irradi-ance data for image light compensation, eliminating the in-terference of ambient light on the data to improve the accuracy and consistency of the data collected in different time periods. The data can be collected at different times of the day to improve accuracy and consistency. During the multispectral image acquisition process, a clear and windless midday weather is selected, the UAV flies autonomously according to the set route and records the images, the multispectral camera lens is vertically downward, and the flight parameters are shown in Table 1.

### 2.3 Data collection plan

The data collection consisted of SPAD of sugar beet canopy and UAV multispectral image data. The SPAD and UAV multispectral image data of sugar beet were collected on 18th June

Table 1. Parameters of UAV multispectral image acquisition.

| Parameter | Parameter values |
|---|---|
| Flight altitude | 12m |
| Flight Speed | 5.4km/h |
| Course overlap ratio | 75% |
| Lateral overlap rate | 75% |
| Spectral type | Blue, Green, Red, Red_edge, Nir |

2022, which is the rapid growth period of leaf clusters of sugar beet. And all da-ta in the collection plan included two different water treatments (normal irrigation and drought stress).

## 2.4 Determination of chlorophyll content

The correlation coefficient between SPAD value and chlorophyll content of sugar beet leaves was significant, which can reflect the high and low levels of chlorophyll content of the crop. The measurement period was the rapid growth period of leaf tufts during the important reproductive period of sugar beet. The relative chlorophyll content of different genotypes was measured simultaneously on the day of the UAV flights by taking five uniformly growing sugar beets from each variety and measuring them using the SPAD-502 Plus chlorophyll meter, which is manufactured by Minolta Camera Company, Japan, and which has been used by many scholars to obtain SPAD data for crops [26, 27]. Measurements were made and SPAD values were recorded at the top, middle and bottom of the canopy of selected beets in the experimental field, and the average of the chlorophyll contents of the three parts of the plant was taken as the canopy SPAD value of the plant, and then the average of the SPAD values of the five plants was calculated as the canopy SPAD value of the variety of beets.

## 2.5 Processing of images

In this study, images were captured by DJI multispectral version P4M UAV, and Pix4Dmapper software (https://pix4d.com/) was used to stitch the acquired raw images, and image correction was performed according to the corresponding image control points on the ground before stitching to generate Digital Orthophoto Map (DOM); then reflectance conversion was performed by whiteboard correction to convert pixel values to reflectance, and finally, reflectance images of all spectra were acquired by ARCGIS software (Version 1). DOM); then the reflectance was converted from pixel values to reflectance by white-board correction to obtain reflectance images of all spectra, and finally, the plot was ex-tracted by ARCGIS software (Version 10.3.1 Esri, USA) (http://www.esri.com/arcgis/about-arcgis) to obtain vectorial surface extraction to obtain the reflectance data of this study plot to provide the data base for the subsequent vegeta-tion index, the specific process is shown in Fig 1.

## 2.6 Selection of vegetation index

Vegetation indices are composed by combining the changes of reflectance of different bands, which can reduce the influence of background soil and other factors on the vegeta-tion spectrum to a certain extent, and improve the accuracy of estimating chlorophyll content. In this paper, a variety of vegetation indices are used, and then the correlation between vegetation indices and SPAD values is combined to preferably select the vegeta-tion indices, and finally the preferred vegetation indices are used to model inversion and prediction of SPAD values. The formula for calculating the vegetation index is shown in Table 2.

## 2.7 Analysis methods

In order to predict the chlorophyll content of sugar beet, this study carried out five types of algorithms, namely K-NearestNeighbor (KNN) [35], Lasso [36], Ran-dom-Forest [37], RidgeCV [38] and Support Vector Machine (SVM) [39], which are the most classical algorithms of machine learning, to study, then the model inversion is performed with the studied data, and finally, through the experimental control of the data, we try to find out the optimal regression learning algorithm that is most suitable for this study, and provide model support for the subsequent data prediction. The specific flowchart of pro-gramme execution is shown in Fig 2.

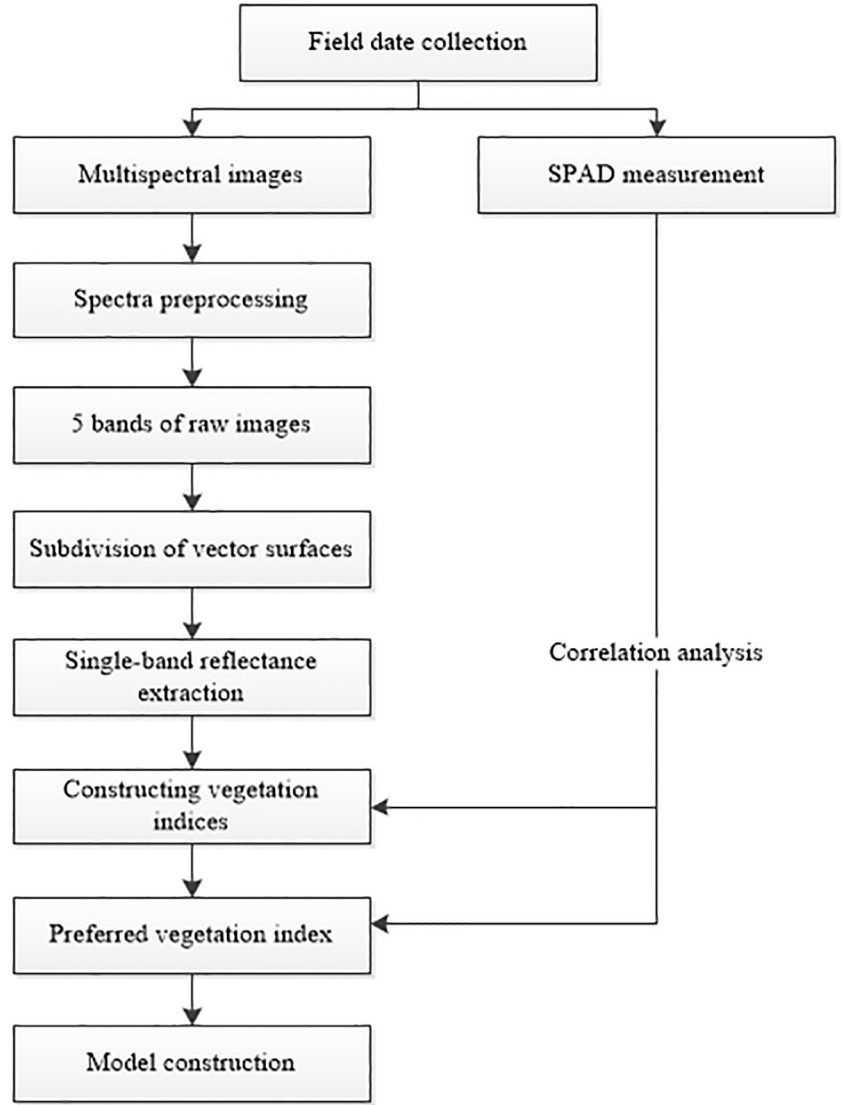

**Fig 1. Flow chart of multispectral image processing.**

**Table 2. Vegetation index and its calculation formula.**

| Vegetation index | formula to calculate | Reference |
|---|---|---|
| NDVI | $NDVI = (R_{Nir} - R_{Red})/(R_{Nir} + R_{Red})$ | [28] |
| GNDVI | $GNDVI = (R_{Nir} - R_{Green})/(R_{Nir} + R_{Green})$ | [29] |
| MSR | $MSR = (R_{Nir}/R_{Red} - 1)/(R_{Nir}/R_{Red} + 1)$ | [30] |
| SR | $SR = R_{Nir}/R_{Red}$ | [31] |
| GCI | $GCI = R_{Nir}/R_{Green} - 1$ | [32] |
| NDREI | $NDREI = (R_{Rededge} - R_{Green})/(R_{Rededge} + R_{Green})$ | [33] |
| OSAVI | $OSAVI = (R_{Nir} - R_{Red})/(R_{Nir} + R_{Red} + 0.16)$ | [34] |

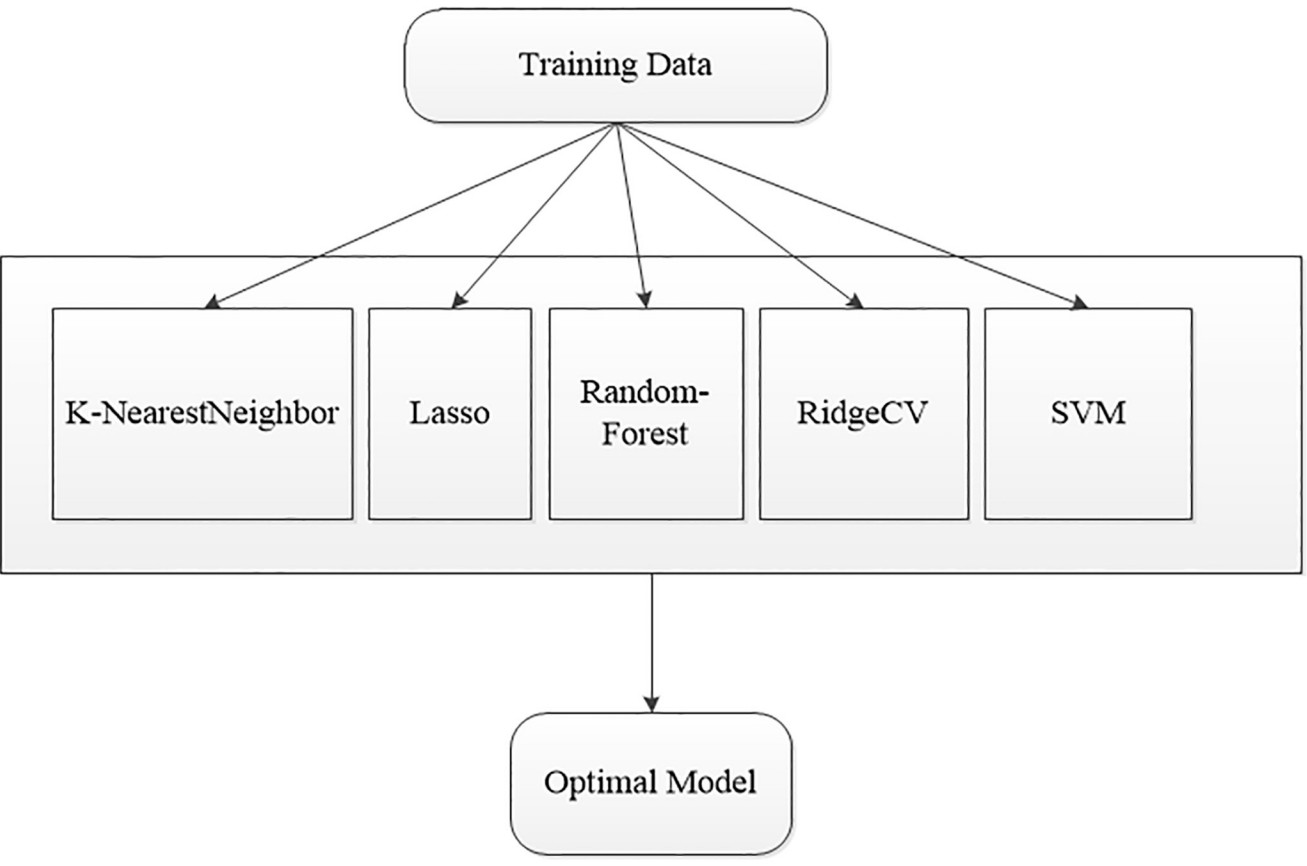

**Fig 2. Flow chart of inverse model of multiple machine learning algorithms.**

## 2.8 Model evaluation indexes

Pearson correlation coefficient R2, root mean square error RMSE and standard root mean square error NRMSE are used as the evaluation indexes of different models, in which the closer the R2 is to 1, the lower the RMSE value indicates that the model's pre-dicted and measured values are in good consistency, and the lower the NRMSE value is, the higher the accuracy of its estimation model and the better the results are, and the mod-el's precision is very high when it is less than 10%, higher when it is between 10% and When it is less than 10%, the model accuracy is high, from 10% to 20%, the accuracy is high, from 20% to 30%, the accuracy is normal, and when it is more than 30%, the accu-racy is poor, and all the above data statistics methods are done using sklearn package in Python language.

## 3 Results

### 3.1 Accuracy evaluation of UAV multispectral image data

The UAV images were subjected to reflectance extraction of different band spectra according to the plots, as shown in Fig 3, the spectral reflectance of different fertility periods were all in the band range of 450–550 nm, and the spectral reflectance curves of different regions showed an increasing trend, and the phenomenon of green wave peaks appeared around 550 nm, which was more consistent with the results of the literature. Between 630nm-670nm it is

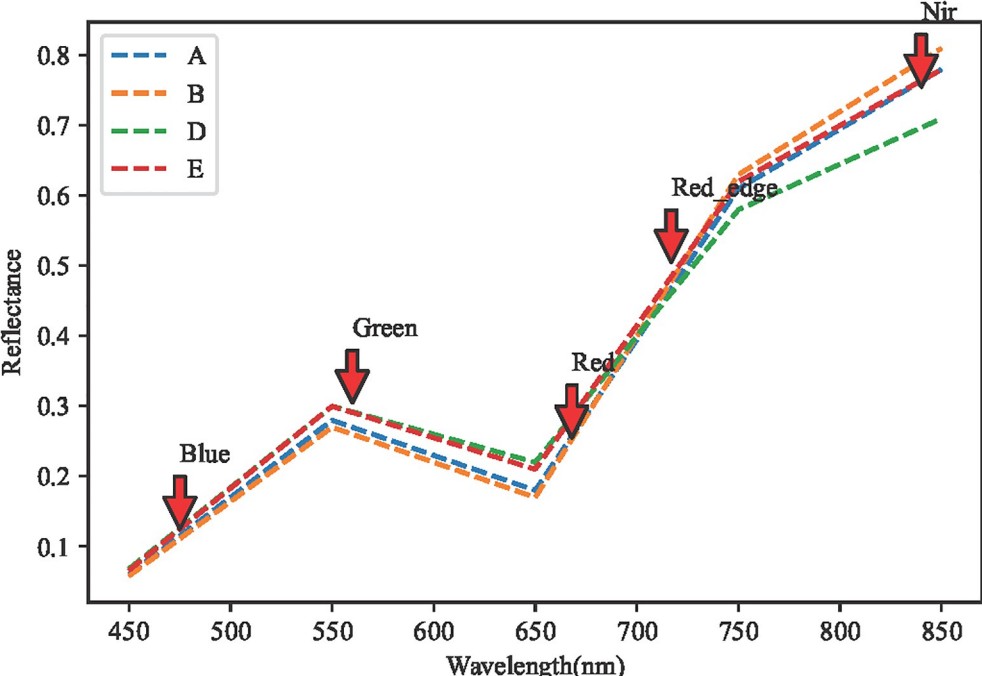

**Fig 3. Spectral reflectance of five bands under water and drought treatments of Manas sugar beet UAVs.** Note: A–B, denote two replicates under normal irrigation; D–E denote two replicates under wa ter limited treatment.

obvious that the position of a red wave valley appears once, and the law of the appearance of this red wave valley is basically the same as that of the appearance of the green wave peak. In the range of 466nm-830nm band, the reflectance of the multispectral data has a high precision, and this result is more consistent with the re-sults of the literature, and the five multispectral bands selected in this paper are in the range of this band, which can be used to estimate the chlorophyll of the canopy of sugar beet.

## 3.2 Distribution of SPAD phenotypes in sugar beet canopy

Canopy SPAD values were obtained for the rapid growth period of sugar beet leaf clusters under different water treatments of normal irrigation and drought stress, respec-tively. SPAD was evaluated by four dimensions, which are mean expressed as ¦Ì, median as median, coeffi-cient of variation as cv, and standard deviation as ¦Ò. From A and B in the Fig 4, it can be seen that the mean value of SPAD under drought stress was distributed in the range of 51.75–53.05, the median was distributed in the range of 51.80–53.13, the distribution of ¦Ò was distributed in the range of 3.64–3.78, and the coefficient of variation of cv was distributed in the range of 6.90%-7.30%. From D and E in the figure, it can be seen that the mean values of SPAD under normal irrigation were distributed in the range of 47.44–50.75, the distribution of median was in the range of 47.40–50.72, the distribution of ¦Ò was in the range of 3.51–3.61, and the coeffi-cient of variation of cv was in the range of 6.90%-7.60%. The coefficient of variation of SPAD of sugar beet under normal irrigation showed a decreasing trend compared to the drought stress data, and also the SPAD of this population under normal irrigation was greater than that under drought stress during the rapid growth period of leaf clusters. In conclusion, the overall data dispersion and the range of variability are still large, and it also indicates that the popula-tion is rich in genetic variation.

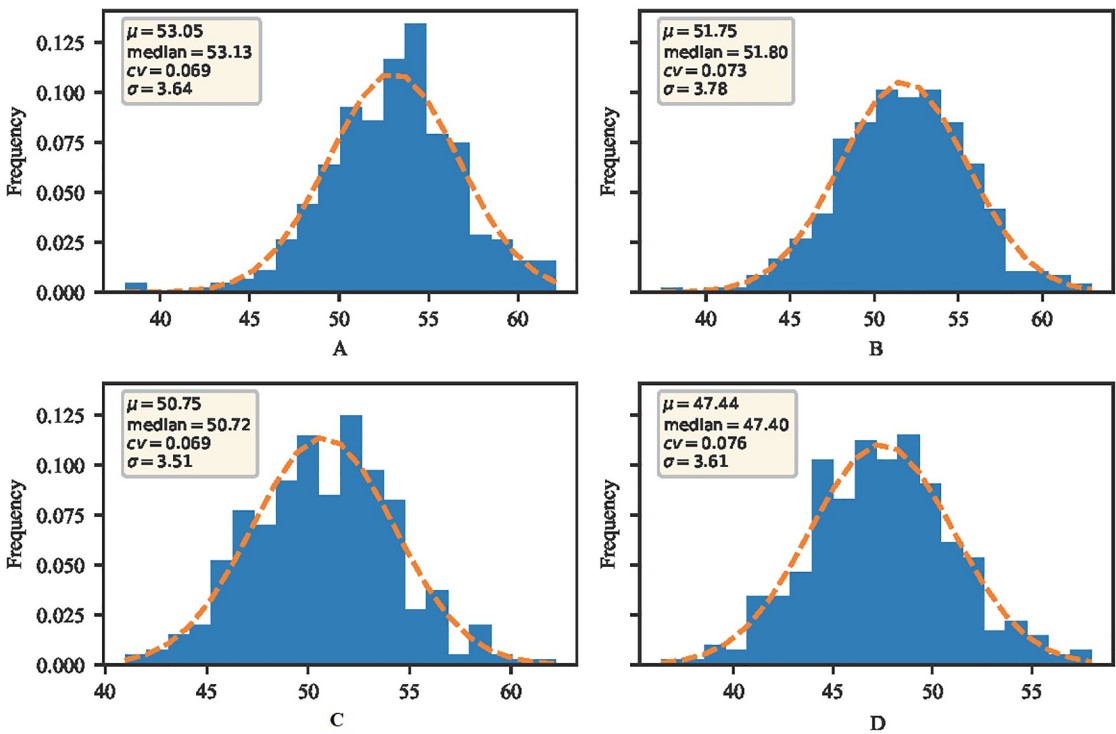

**Fig 4. Distribution of SPAD during the rapid growth period of Manas beet leaf clusters.**

### 3.3 Correlation analysis between SPAD and vegetation index of sugar beet canopy

Correlation analysis of canopy SPAD values during the rapid growth period of sugar beet leafy bush with spectral parameters corresponding to the fertility period was carried out and the results are shown in Fig 5. From Fig 5, it can be seen that most of the spectral vegetation indices under normal irrigation during the rapid growth period of sugar beet leafy bush reached highly significant levels ($p<0.001$). The correlation coeffi-cients of NDVI and SR vegetation indices reached 0.60, and the others were above 0.50; SPAD also reached highly significant levels ($p<0.001$) with the corresponding vegetation indices under drought stress; the correlation coefficients of SPAD and NDVI vegetation indices reached -0.60, and the other vegetation indices reached -0.55 or above. Combining the results of correlation analyses between SPAD and various vegetation indices during the rapid growth period of leaf clusters in the important fertility period of sugar beet, the correlations of the selected vegetation indices reached a significant level, and the next step can be the inversion of model prediction for SPAD of sugar beet.

### 3.4 Evaluation of the accuracy of the prediction model

Five machine learning algorithms, namely KNN, Lasso, Random-Forest, RidgeCV and SVM, were used to invert the model under two water treatments for the SPAD values of the canopy during the rapid growth period of the leaf clusters of sugar beets, respec-tively, and the results are shown in Fig 6. The correlation coefficients of SPAD predic-tion using the RNN model under normal irrigation were R2 = 0.481, Rmse = 2.5, and Re = 0.6%; the correlation coeffi-cients R2 for SPAD prediction using the Lasso model were 0.630, Rmse = 2.12 and Re = 0.80%; the correlation coefficients R2 for SPAD prediction using the Random-Frost model were

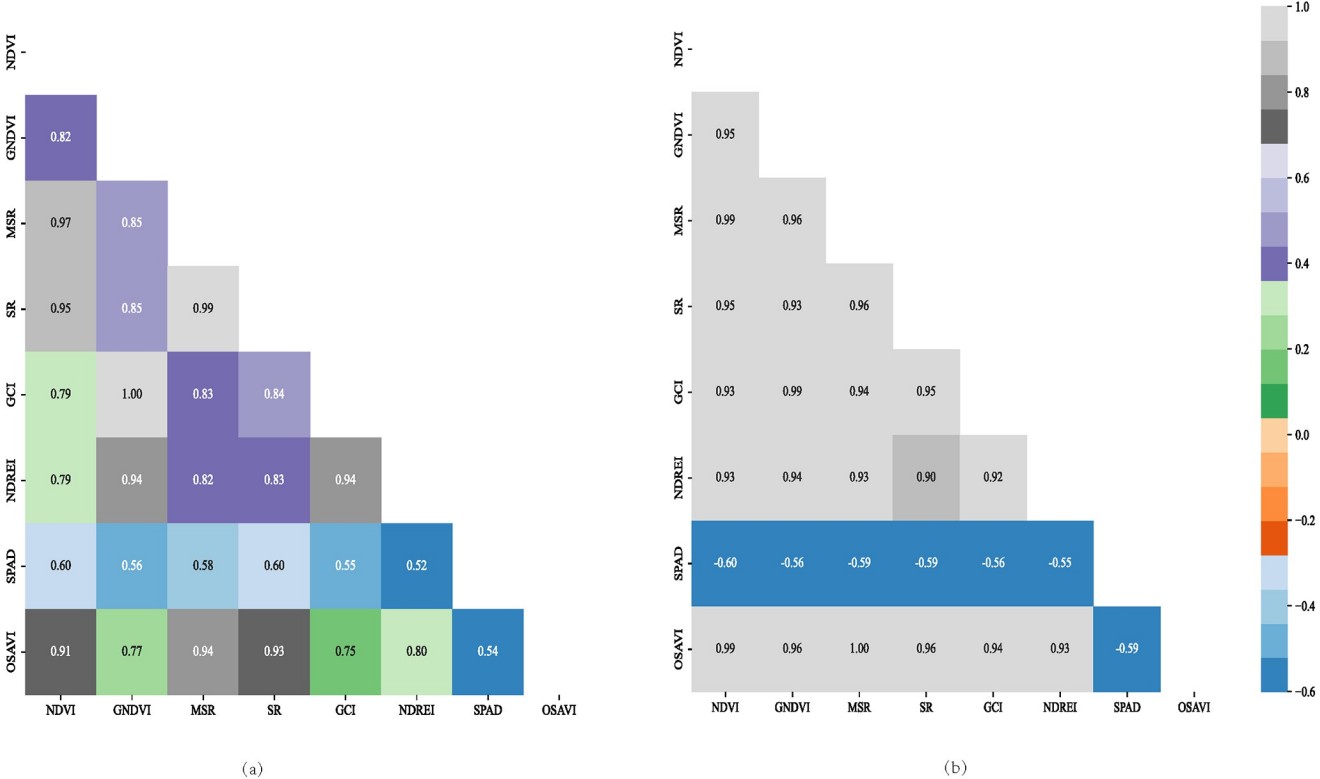

**Fig 5. Correlation of different vegetation indices with SPAD of sugar beet during rapid growth period of leaf clusters.** (a) Normal treatment during rapid growth period of leaf clusters of Manas sugar beet. (b) Water treatment during rapid growth period of leaf clusters of Manas sugar beet.

0.606, Rmse = 2.19 and Re = 0.3%; and the correlation coefficients $R^2$ for SPAD prediction using the RidgeCV model were 0.3%. The correlation coefficients for SPAD prediction using the RidgeCV model $R^2$ = 0.606, Rmse = 2.19 and Re = 0.3%; and the correlation coefficients for SPAD prediction using the SVM model $R^2$ = 0.635, Rmse = 2.13 and Re = 0.8%. Under drought stress, the correlation coefficients for SPAD prediction using the RNN model $R^2$ = 0.518, Rmse = 2.89 and Re = 0.01. The correlation coefficients for SPAD prediction using the Lasso model $R^2$ = 0.610, Rmse = 2.7 and Re = 0.1%; and the correlation coefficients for SPAD prediction using the Random-Frost model $R^2$ = 0.512, Rmse = 2.91, and Re = 0.4%; the correlation coefficient $R^2$ for SPAD prediction using the Ridgecv model was 0.512, Rmse = 2.91 and Re = 0.4%; and the correlation coefficient $R^2$ for SPAD prediction using the SVM model was 0.609, Rmse = 2.71 and Re = 0.1%.

## 4 Discussion

### 4.1 Effects of drought and water treatments on chlorophyll

In drought environments, plants have evolved a series of mechanisms for self-protection and adaptation to and resistance to unfavourable environmental stresses, and their phenotypic traits are significantly altered to minimize the effects of the adverse environment on their growth and development. At the same time, drought stress has complex and multifaceted effects on the population structure and physiology of crops [40]. There are more studies on the effects of drought on the above-ground parts of sugar beet, and it has been shown that there is

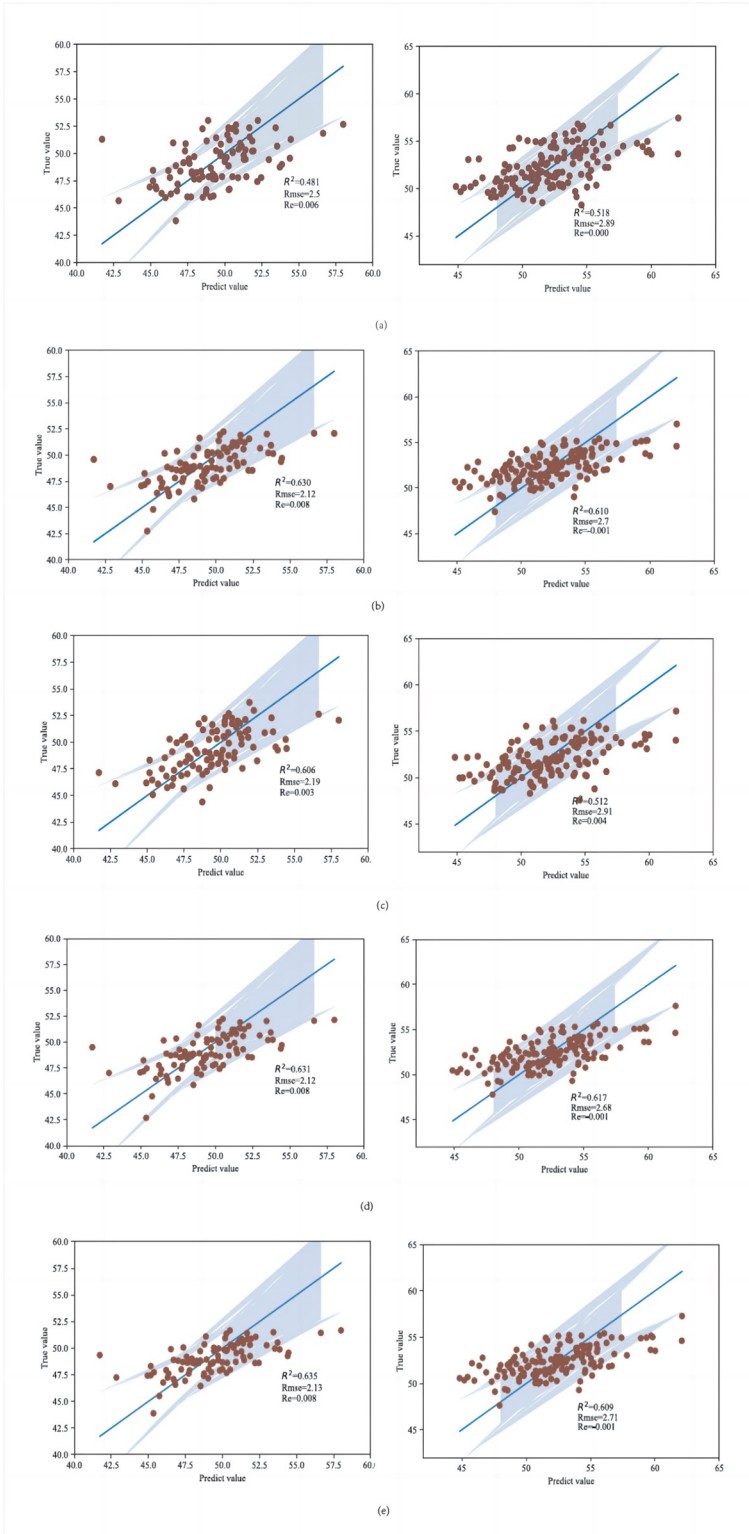

**Fig 6. Prediction results of five machine learning models for rapid growth period of SPAD leaf clusters in sugar beet canopy under normal irrigation and drought stress.** (a) KNN model prediction results for rapid growth period of SPAD leaf clusters in sugar beet canopy under normal irrigation and drought stress. (b) Lasso model prediction results for rapid growth period of SPAD leaf clusters in sugar beet canopy under normal irrigation and drought stress. (c) Random-Forest model predictions of rapid growth period of SPAD leaf clusters in sugar beet canopy under normal

irrigation and drought stress. (d) RidgeCV model predictions of rapid growth period of SPAD leaf clusters in sugar beet canopy under normal irrigation and drought stress. (e) SVM model predictions of rapid growth period of SPAD leaf clusters in sugar beet canopy under normal irrigation and drought stress.

a linear positive correlation between yield and leaf ar-ea of sugar beet under drought stress [41]. The production of reactive oxygen species (ROS) in the organelles under drought is one of the main factors affecting crop growth and de-velopment, and during photosynthesis, the lack of energy dissipated by excited chloro-phyll leads to the formation of chlorophyll triplet states and the reaction with triplet oxy-gen to generate highly reactive oxygen species, thus suggesting that chlorophyll content is closely associated with both drought tolerance and growth and development of sugar beet.

In this study, the analysis of chlorophyll content of sugar beet canopy showed that the results indicated that normal irrigation conditions highly significantly reduced the chlorophyll content of sugar beet canopy leaf clusters during the rapid growth period than drought stress. In addition, chlorophyll is the most important pigment for photosynthesis, which affects the physiological and biochemical processes in crops under drought stress. Drought stress causes the reactive oxygen species produced by plants to damage cell membranes, hindering chloro-phyll synthesis and accelerating degradation, thus reducing chlorophyll content. At the same time, previous researchers have shown that in the early stage of stress, SOD activity and POD activity in sugar beet roots showed an increasing trend to protect sugar beet from drought stress, but in the later stage, SOD activity and POD activity gradually decreased with the inten-sification of the degree of stress until the end of the stress [42]. Drought stress promotes the increase of proline and malondialde-hyde in crop roots, and the promotion is stronger with increasing drought [43]. The drought stress situation increased the antioxidant enzyme activity of the crop, exacerbated the degree of lipid peroxidation in the crop root system membrane, and accumulated os-moregulatory substances to a certain extent [44, 45]. Meanwhile, previous studies also showed that the canopy SPAD value of sugar beet under drought stress showed a de-creasing trend, and sugar beet varieties with higher SPAD values under stress had higher yields and better drought tolerance [46]. These findings are consistent with the results of this paper that the SPAD values of sugar beet canopy under drought stress showed a de-creasing trend.

### 4.2 Generalisation of the SPAD inversion models

The current UAV multispectral with high spectral resolution and flexible manoeu-vrability plays an important role in crop high-throughput phenotyping research. In this paper, a UAV-mounted multispectral camera was used to collect ground image data and estimate the ground SPAD content [47]. The reflectance extraction of the image data re-vealed that the spectral reflectance curves of the different fertility periods studied in this paper and can be seen in the phenomenon of green light wave peaks at a wavelength of about 550 nm, which is more consis-tent with the results of the literature [48, 49]. Between 630nm-670nm it is obvious that the position of a red wave valley appeared once, and the law of the appearance of this red wave val-ley and the law of the appearance of the green wave peak is basically the same. In the range of 466nm-830nm band, the reflectance of multispectral data has a high accuracy, and this result is more consistent with the results of Aasen et al. [48] and Zhao et al. [50]. A single vegetation index does not adequately reflect crop growth, but too many veg-etation indices as input parameters to the model [51, 52] will lead to an increase in the complexity of the model. There-fore, before the model construction, different optimal vege-tation indices were obtained

through the correlation between vegetation indices and SPAD. In terms of the prediction model of SPAD, this paper investigated the SPAD prediction model by five machine learning algorithmic models under normal irrigation and water restriction treatments for the rapid growth period of leaf clusters of sugar beet, respectively. It was found that the prediction models for different water treatments at different fertility periods were different, but the model with higher correlation between predicted and true values under different water treatments was the SVM model, which embodied a strong advantage among all models in terms of goodness-of-fit and accuracy.

## 4.3 Conclusions

In this study, a method utilizing unmanned aerial vehicle (UAV) remote sensing im-agery is developed for estimating the SPAD (Soil Plant Analysis Development) of sugar beet canopies. The approach integrates field-measured data and aerial UAV remote sens-ing data, incorporating the selection of vegetation indices and the assessment of the corre-lation between SPAD and vegetation indices. Five machine learning algorithms, namely KNN (K-Nearest Neighbors), Lasso, Random Forest, Ridgecv, and SVM (Support Vector Machine), are employed for model inversion. Under normal irrigation conditions, the cor-relation coefficients ($R^2$) between predicted and test values are 0.635, RMSE (Root Mean Square Error) is 2.13, and RE (Relative Error) is 0.80%. Under drought stress, the corre-sponding values are $R^2$ = 0.609, RMSE = 2.71, and RE = 0.10%. The results of this study demonstrate that constructing predic-tion models using the SVM algorithm under different water treatments can effectively invert the chlorophyll content of sugar beet canopies with varying growth conditions.

## Supporting information

**S1 Data.**
(XLSX)

## Author Contributions

**Conceptualization:** Weishi Gao, Liming Zhang.

**Data curation:** Wei Wang, Jikun Song, Hao Wu.

**Formal analysis:** WanYing Zeng, Sizhong Li.

**Investigation:** Wei Wang, Jikun Song, Hao Wu.

**Methodology:** Wei Wang, Jikun Song, Hao Wu.

**Project administration:** Weishi Gao, Liming Zhang.

**Resources:** Weishi Gao, Liming Zhang.

**Supervision:** Liming Zhang.

**Writing – original draft:** Weishi Gao, WanYing Zeng, Sizhong Li, Liming Zhang.

**Writing – review & editing:** Weishi Gao, WanYing Zeng, Sizhong Li, Liming Zhang.

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
