## [Decision Letter · Decision Letter 0]

3 Jan 2024

PONE-D-23-33769Remote sensing estimation of sugar beet SPAD based on unmanned aerial vehicle multispectral imageryPLOS ONE

Dear Dr. Li,

Thank you for submitting your manuscript to PLOS ONE. After careful consideration, we feel that it has merit but does not fully meet PLOS ONE’s publication criteria as it currently stands. Therefore, we invite you to submit a revised version of the manuscript that addresses the points raised during the review process.

We look forward to receiving your revised manuscript.

Kind regards,

Habib Ali, PhD

Academic Editor

PLOS ONE

“This research was funded by Key Research and development task special project of Xin-jiang(2022B02002); National Sugar Industry Technology System Project (CARS-170108); Henan Provincial Science and Technology Tackling Project(232102111132).”

5. We note that your Data Availability Statement is currently as follows: [All relevant data are within the manuscript and its Supporting Information files.]

6. PLOS requires an ORCID iD for the corresponding author in Editorial Manager on papers submitted after December 6th, 2016. Please ensure that you have an ORCID iD and that it is validated in Editorial Manager. To do this, go to ‘Update my Information’ (in the upper left-hand corner of the main menu), and click on the Fetch/Validate link next to the ORCID field. This will take you to the ORCID site and allow you to create a new iD or authenticate a pre-existing iD in Editorial Manager. Please see the following video for instructions on linking an ORCID iD to your Editorial Manager account: https://www.youtube.com/watch?v=_xcclfuvtxQ.

7. We note that Figure 1 in your submission contain copyrighted images. All PLOS content is published under the Creative Commons Attribution License (CC BY 4.0), which means that the manuscript, images, and Supporting Information files will be freely available online, and any third party is permitted to access, download, copy, distribute, and use these materials in any way, even commercially, with proper attribution. For more information, see our copyright guidelines: http://journals.plos.org/plosone/s/licenses-and-copyright.

Additional Editor Comments:

The use of language could be more precise and concise. Consider avoiding unnecessary words to enhance clarity. Ensure consistent use of tense throughout the paper for a smoother flow of ideas. The paper would benefit from providing more context and background information to help readers better understand the research problem. author should Work on transitions between paragraphs to enhance the overall coherence and make the paper more reader-friendly.There are typographical errors present. Consider thorough proofreading to eliminate spelling mistakes and improve the overall professionalism of the paper.These comments are intended to guide improvements in language, grammar, and overall paper structure. It is advisable to carefully review and revise the paper with these considerations in mind.

Reviewers' comments:

Reviewer's Responses to Questions

**Comments to the Author**

1. Is the manuscript technically sound, and do the data support the conclusions?

Reviewer #1: Yes

Reviewer #2: Yes

2. Has the statistical analysis been performed appropriately and rigorously? 

Reviewer #1: Yes

Reviewer #2: Yes

3. Have the authors made all data underlying the findings in their manuscript fully available?

Reviewer #1: Yes

Reviewer #2: Yes

4. Is the manuscript presented in an intelligible fashion and written in standard English?

Reviewer #1: Yes

Reviewer #2: Yes

5. Review Comments to the Author

Reviewer #1: I appreciate the Editor to give me a chance to review an interesting and valuable paper. I found some merits in the both methodology and results. However, I have also some concerns on the manuscript. If the author(s) address to the comments, I’ll recommend this paper entitled “Remote sensing estimation of sugar beet SPAD based on un-manned aerial vehicle multispectral imagery” for publication with minor revision:

1.I noticed that manuscript is very good but authors not corrected cited the reference, please correct them and use latest reference.

2.Double check the reference and arrange according to the journal style.

3.L52: You not added any reference here. Add this “a new solution for the canopy traits of crops (Tian et al., 2020) and this study focuses with the SPAD prediction of sugar beet based on multispectral images from UAV remote sensing (Tian et al., 2019).” Add references according to the journal like [5] and [6].

Tian H, Huang N, Niu Z, Qin Y, Pei J, Wang J. Mapping winter crops in china with multi-source satellite imagery and phenology-based algorithm. Remote Sens. 2019; 11: 820. https://doi.org/10.3390/rs11070820

Tian H, Pei J, Huang J, Li X, Wang J, Zhou B, Qin Y, Wang L. Garlic and winter wheat identification based on active and passive satellite imagery and the google earth engine in northern china. Remote Sens, 2020; 12, 3539. https://doi.org/10.3390/rs12213539 “

4.L56: You not add any reference after “rapidly in both breadth and depth.” Add these as “UAV remote sensing of agriculture has been developing rapidly in both breadth and depth (Zhao et al., 2023; Cheng et al., 2023). Add references according to the journal like [7, 8].

Zhao K, Jia Z, Jia F, Shao H. Multi-scale integrated deep self-attention network for predicting remaining useful life of aero-engine. Eng Appl Artif Intell, 2023; 120, 105860. https://doi.org/10.1016/j.engappai.2023.105860

Cheng Y, Lan S, Fan X, Tjahjadi T, Jin S, Cao L. A dual-branch weakly supervised learning based network for accurate mapping of woody vegetation from remote sensing images. Int J Appl Earth Obs Geoinf, 2023; 124, 103499. https://doi.org/10.1016/j.jag.2023.103499

5.L58: You not added any reference, Please add like “they have become the favoured object in the majority of research (Zhou et al., 2021; Zhuo et al., 2022).” Add references according to the journal like [9, 10].

Zhou G, Li W, Zhou X, Tan Y, Lin G, Li X, Deng R. An innovative echo detection system with STM32 gated and PMT adjustable gain for airborne LiDAR. Int J Remote Sens, 2021; 42:24, 9187-9211. https://doi.org/10.1080/01431161.2021.1975844

Zhuo Z, Du L, Lu X, Chen J, Cao Z. Smoothed Lv Distribution Based Three-Dimensional Imaging for Spinning Space Debris. IEEE Trans Geosci Remote Sens, 2022; 60, 1-13. https://doi.org/10.1109/TGRS.2022.3174677

6.L60: You not added any reference, add like “spectral sensors, multispectral sensors, etc (Zhang et al., 2023; Zhao et al., 2023).” Add references according to the journal like [11, 12].

Zhang Y, Li S, Wang S, Wang X, Duan H. Distributed bearing-based formation maneuver control of fixed-wing UAVs by finite-time orientation estimation. Aerosp Sci Technol, 2023; 136, 108241. https://doi.org/10.1016/j.ast.2023.108241

Zhao J, Gao F, Jia W, Yuan W, Jin W. Integrated Sensing and Communications for UAV Communications with Jittering Effect. IEEE Wirel Commun Lett, 2023; 12, 758-762. https://doi.org/10.1109/LWC.2023.3243590

7.L62-63: “At present, domestic and foreign scholars use hyperspectral for the estimation of vegetation chlorophyll content”. It’s not correct. Please remove at present, domestic and foreign scholar, write just many researchers. Correct this sentence.

8.L68: You write “Bendig et al[7] predicted the biomass of crops by UAV and obtained better results.” Give space between al and [7] and also give another reference as “Bendig et al. [7] and Dai et al. (2023) predicted the biomass of crops by UAV and obtained better results.” Add references according to the journal like Bendig et al. [14] and Dai et al. [15].

Dai X, Xiao Z, Jiang H, Lui JCS. UAV-Assisted Task Offloading in Vehicular Edge Computing Networks. IEEE Trans Mob Comput. 2023; 21, 3536-3550. https://doi.org/10.1109/TMC.2023.3259394

9.L71: You not add any reference here. Add reference as “remote sensing in precision agriculture (Zhou et al., 2021; Liu et al., 2022). Add references according to the journal like [17,18]

Zhou G, Deng R, Zhou X, Long S, Li W, Lin G, Li X. Gaussian Inflection Point Selection for LiDAR Hidden Echo Signal Decomposition. IEEE Geosci Remote Sens Lett. 2021; 19, 1-5. https://doi.org/10.1109/LGRS.2021.3107438

Liu H, Li J, Meng X, Zhou B, Fang G, Spencer B.F. Discrimination Between Dry and Water Ices by Full Polarimetric Radar: Implications for China’s First Martian Exploration. IEEE Trans Geosci Remote Sens, 2022; 61, 1-11. https://doi.org/10.1109/TGRS.2022.3228684

10.L73: You not add any reference. Please add reference as “unknown data based on established models and input parameters (Lin et al., 2022; She et al., 2022).” Add references according to the journal like [20, 21]

Lin Z, Wang H, Li S. Pavement anomaly detection based on transformer and self-supervised learning. Autom Constr. 2022; 143, 104544. https://doi.org/10.1016/j.autcon.2022.104544

She Q, Hu R, Xu J, Liu M, Xu K, Huang H. Learning High-DOF Reaching-and-Grasping via Dynamic Representation of Gripper-Object Interaction. ACM Trans Graph. 2022; 41(4). https://doi.org/10.1145/3528223.3530091

11.L75-76: You not added any reference, please add reference as “but also in unstructured and non-linear domains (Zhou et al., 2021; Zhang et al., 2021). Add references according to the journal like [22, 23]

Zhou G, Zhang R, Huang S. Generalized Buffering Algorithm. IEEE access, 2021; 9, 27140-27157. https://doi.org/10.1109/ACCESS.2021.3057719

Zhang J, Zhu C, Zheng L, Xu K. ROSEFusion: random optimization for online dense reconstruction under fast camera motion. ACM Trans Graph. 2021; 40, 1-17. https://doi.org/10.1145/3450626.3459676

12.L78: Reference [8-10] is not suitable, remove them and add following reference instead to these at this place.

Mao Y, Zhu Y, Tang Z, Chen Z. A Novel Airspace Planning Algorithm for Cooperative Target Localization. Electronics. 2022; 11, 2950. https://doi.org/10.3390/electronics11182950

Yin L, Wang L, Li J, Lu S, Tian J, Yin Z, Liu S, Zheng W. YOLOV4_CSPBi: Enhanced Land Target Detection Model. Land, 2023; 12, 1813. https://doi.org/10.3390/land12091813

13.L203: Please don’t use any reference in the results section. Remove all from the results.

14.L330: Add this reference here after the “estimate the ground SPAD content (Cao et al., 201)” Add references according to the journal like [31]

Cao B, Li M, Liu X, Zhao J, Cao W, Lv Z. Many-Objective Deployment Optimization for a Drone-Assisted Camera Network. IEEE Trans Netw Sci Eng, 2021; 8, 2756-2764. https://doi.org/10.1109/TNSE.2021.3057915

15.L333: “which is more consistent with the results of the literature [26]” You add only 1 reference. Add 1 more as I suggested and given following “Wang et al., 2022”.

Wang H, Zhang X, Jiang S. A Laboratory and Field Universal Estimation Method for Tire–Pavement Interaction Noise (TPIN) Based on 3D Image Technology. Sustainability, 2022; 14, 12066. https://doi.org/10.3390/su141912066

16.L338: Add following reference as well with the “result is more consistent with the results of literature [26]” Also remove the word literature. Just write the “results of the Aasen et al. [26] and Zhao et al. [7]”.

Zhao F, Wu H, Zhu S, Zeng H, Zhao Z, Yang X, Zhang S. Material stock analysis of urban road from nighttime light data based on a bottom-up approach. Environ Res, 2023; 228, 115902. https://doi.org/10.1016/j.envres.2023.115902

17.L340-341: “model will lead to an increase in the complexity of the model.” Add following reference here as “model will lead to an increase in the complexity of the model (Zheng et al., 2023; Yin et al., 2023). Add references according to the journal like [20, 21].

Zheng H, Fan X, Bo W, Yang X, Tjahjadi T, Jin S. A Multiscale Point-Supervised Network for Counting Maize Tassels in the Wild. Plant Phenomics. 2023; 5, 100. https://doi.org/10.34133/plantphenomics.0100

Yin L, Wang L, Li T, Lu S, Tian J, Yin Z., et al. U-Net-LSTM: Time Series-Enhanced Lake Boundary Prediction Model. Land, 2023; 12, 1859. https://doi.org/10.3390/land12101859

18.Don’t repeat results in discussion section.

19.In conclusion you just make repetition of similar information as given in the start of conclusion. Make it clear.

20.Need to focus on the findings according to result. What are recommendations?

21.Double check that all references are cited within the text

Overall paper is very good. I recommend the minor revision.

Reviewer #2: Dear Editor,

Gao et al. attempted to describe remote sensing estimation of sugar beet SPAD based on unmanned aerial vehicle multispectral imagery; it is a nice piece of work, I would like it to be published, however there are some points, cited below, which need author’s attention.

1.In lines 37-39 taxonomical details for the studied plant species are erroneous. Sugar beet (scientific name: Beta vulgaris) belongs to Amaranthaceae family and genus Beta and not as it is cited [genus Sugar Beet, family Quinoa]. The manuscript do not cite the scientific name of the studied plant species.

2.In lines 92-94 it is cited that [300 copies of domestic and foreign staple varieties were selected as the participating materials…]. There is no available information a) for their genetic and/or physiological background b) for their resistance to stress factor (drought). Further explanations are needed due to the fact that outcome UAV spectral data are not linked with known informative basis of the crop.

In Result section lines: 226-229 “In conclusion, the overall data dispersion and the range of variability are still large, and it also indicates that the population is rich in genetic variation. Genetic variation should be standardized, defined and stated prior to the experiment; authors received SPAD response variation in stressed and non-stressed conditions which cannot provide reverse statements for optimized materials experimentation.

3.Drone platforms and flight setting are well described; however no geographical spatial / positioning information was provided for the scanned area.

4.Authors provide extensive discussion on causes of chlorophyll decline due to drought (section 4.1-Discussion); however, this information is not taken into account on experimental planning and outcome results. Core aim of the manuscript is to introduce UAV remote sensing crop phenotyping in machine learning (as it is indicated in lines 79-87) but there is no categorization and or validation of drought-chrolophyll decline causes prior data being introduced to algorithmic – machine learning analysis phase.

5.Some inconsistency typing problems found citation’s section format (capital letters text etc).

In conclusion, according to my opinion the manuscript has some interest viewpoints in terms of UAV spectral scans on sugar beet plantations introducing outcome data for machine learning processes; if some clarifications /changes take place I would be glad to see it published as one more step in machine learning and UAV spectral analysis of crops.

6. PLOS authors have the option to publish the peer review history of their article (what does this mean?). If published, this will include your full peer review and any attached files.

Reviewer #1: **Yes: **Muhammad Saqlain Zaheer

Reviewer #2: No

---

## [Author Response · Author response to Decision Letter 0]

18 Feb 2024

Response Letter of PONE-D-23-33769

Dear Editor, and the reviewers, 

 This is the response letter for paper PONE-D-23-33769, entitled “Remote sensing estimation of sugar beet SPAD based on unmanned aerial vehicle multispectral imagery”. We greatly thank the editors and the reviewers for their efforts and their constructive comments that help us improve this study. This letter details our revision in response to their comments.

Response to Reviewer #1

1.I noticed that manuscript is very good but authors not corrected cited the reference, please correct them and use latest reference.

2.Double check the reference and arrange according to the journal style.

3.L52: You not added any reference here. Add this “a new solution for the canopy traits of crops (Tian et al., 2020) and this study focuses with the SPAD prediction of sugar beet based on multispectral images from UAV remote sensing (Tian et al., 2019).” Add references according to the journal like [5] and [6].

Tian H, Huang N, Niu Z, Qin Y, Pei J, Wang J. Mapping winter crops in china with multi-source satellite imagery and phenology-based algorithm. Remote Sens. 2019; 11: 820. https://doi.org/10.3390/rs11070820

Tian H, Pei J, Huang J, Li X, Wang J, Zhou B, Qin Y, Wang L. Garlic and winter wheat identification based on active and passive satellite imagery and the google earth engine in northern china. Remote Sens, 2020; 12, 3539. https://doi.org/10.3390/rs12213539 “

4.L56: You not add any reference after “rapidly in both breadth and depth.” Add these as “UAV remote sensing of agriculture has been developing rapidly in both breadth and depth (Zhao et al., 2023; Cheng et al., 2023). Add references according to the journal like [7, 8].

Zhao K, Jia Z, Jia F, Shao H. Multi-scale integrated deep self-attention network for predicting remaining useful life of aero-engine. Eng Appl Artif Intell, 2023; 120, 105860. https://doi.org/10.1016/j.engappai.2023.105860

Cheng Y, Lan S, Fan X, Tjahjadi T, Jin S, Cao L. A dual-branch weakly supervised learning based network for accurate mapping of woody vegetation from remote sensing images. Int J Appl Earth Obs Geoinf, 2023; 124, 103499. https://doi.org/10.1016/j.jag.2023.103499

5.L58: You not added any reference, Please add like “they have become the favoured object in the majority of research (Zhou et al., 2021; Zhuo et al., 2022).” Add references according to the journal like [9, 10].

Zhou G, Li W, Zhou X, Tan Y, Lin G, Li X, Deng R. An innovative echo detection system with STM32 gated and PMT adjustable gain for airborne LiDAR. Int J Remote Sens, 2021; 42:24, 9187-9211. https://doi.org/10.1080/01431161.2021.1975844

Zhuo Z, Du L, Lu X, Chen J, Cao Z. Smoothed Lv Distribution Based Three-Dimensional Imaging for Spinning Space Debris. IEEE Trans Geosci Remote Sens, 2022; 60, 1-13. https://doi.org/10.1109/TGRS.2022.3174677

6.L60: You not added any reference, add like “spectral sensors, multispectral sensors, etc (Zhang et al., 2023; Zhao et al., 2023).” Add references according to the journal like [11, 12].

Zhang Y, Li S, Wang S, Wang X, Duan H. Distributed bearing-based formation maneuver control of fixed-wing UAVs by finite-time orientation estimation. Aerosp Sci Technol, 2023; 136, 108241. https://doi.org/10.1016/j.ast.2023.108241

Zhao J, Gao F, Jia W, Yuan W, Jin W. Integrated Sensing and Communications for UAV Communications with Jittering Effect. IEEE Wirel Commun Lett, 2023; 12, 758-762. https://doi.org/10.1109/LWC.2023.3243590

7.L62-63: “At present, domestic and foreign scholars use hyperspectral for the estimation of vegetation chlorophyll content”. It’s not correct. Please remove at present, domestic and foreign scholar, write just many researchers. Correct this sentence.

8.L68: You write “Bendig et al[7] predicted the biomass of crops by UAV and obtained better results.” Give space between al and [7] and also give another reference as “Bendig et al. [7] and Dai et al. (2023) predicted the biomass of crops by UAV and obtained better results.” Add references according to the journal like Bendig et al. [14] and Dai et al. [15].

Dai X, Xiao Z, Jiang H, Lui JCS. UAV-Assisted Task Offloading in Vehicular Edge Computing Networks. IEEE Trans Mob Comput. 2023; 21, 3536-3550. https://doi.org/10.1109/TMC.2023.3259394

9.L71: You not add any reference here. Add reference as “remote sensing in precision agriculture (Zhou et al., 2021; Liu et al., 2022). Add references according to the journal like [17,18]

Zhou G, Deng R, Zhou X, Long S, Li W, Lin G, Li X. Gaussian Inflection Point Selection for LiDAR Hidden Echo Signal Decomposition. IEEE Geosci Remote Sens Lett. 2021; 19, 1-5. https://doi.org/10.1109/LGRS.2021.3107438

Liu H, Li J, Meng X, Zhou B, Fang G, Spencer B.F. Discrimination Between Dry and Water Ices by Full Polarimetric Radar: Implications for China’s First Martian Exploration. IEEE Trans Geosci Remote Sens, 2022; 61, 1-11. https://doi.org/10.1109/TGRS.2022.3228684

10.L73: You not add any reference. Please add reference as “unknown data based on established models and input parameters (Lin et al., 2022; She et al., 2022).” Add references according to the journal like [20, 21]

Lin Z, Wang H, Li S. Pavement anomaly detection based on transformer and self-supervised learning. Autom Constr. 2022; 143, 104544. https://doi.org/10.1016/j.autcon.2022.104544

She Q, Hu R, Xu J, Liu M, Xu K, Huang H. Learning High-DOF Reaching-and-Grasping via Dynamic Representation of Gripper-Object Interaction. ACM Trans Graph. 2022; 41(4). https://doi.org/10.1145/3528223.3530091

11.L75-76: You not added any reference, please add reference as “but also in unstructured and non-linear domains (Zhou et al., 2021; Zhang et al., 2021). Add references according to the journal like [22, 23]

Zhou G, Zhang R, Huang S. Generalized Buffering Algorithm. IEEE access, 2021; 9, 27140-27157. https://doi.org/10.1109/ACCESS.2021.3057719

Zhang J, Zhu C, Zheng L, Xu K. ROSEFusion: random optimization for online dense reconstruction under fast camera motion. ACM Trans Graph. 2021; 40, 1-17. https://doi.org/10.1145/3450626.3459676

12.L78: Reference [8-10] is not suitable, remove them and add following reference instead to these at this place.

Mao Y, Zhu Y, Tang Z, Chen Z. A Novel Airspace Planning Algorithm for Cooperative Target Localization. Electronics. 2022; 11, 2950. https://doi.org/10.3390/electronics11182950

Yin L, Wang L, Li J, Lu S, Tian J, Yin Z, Liu S, Zheng W. YOLOV4_CSPBi: Enhanced Land Target Detection Model. Land, 2023; 12, 1813. https://doi.org/10.3390/land12091813

13.L203: Please don’t use any reference in the results section. Remove all from the results.

14.L330: Add this reference here after the “estimate the ground SPAD content (Cao et al., 201)” Add references according to the journal like [31]

Cao B, Li M, Liu X, Zhao J, Cao W, Lv Z. Many-Objective Deployment Optimization for a Drone-Assisted Camera Network. IEEE Trans Netw Sci Eng, 2021; 8, 2756-2764. https://doi.org/10.1109/TNSE.2021.3057915

15.L333: “which is more consistent with the results of the literature [26]” You add only 1 reference. Add 1 more as I suggested and given following “Wang et al., 2022”.

Wang H, Zhang X, Jiang S. A Laboratory and Field Universal Estimation Method for Tire–Pavement Interaction Noise (TPIN) Based on 3D Image Technology. Sustainability, 2022; 14, 12066. https://doi.org/10.3390/su141912066

16.L338: Add following reference as well with the “result is more consistent with the results of literature [26]” Also remove the word literature. Just write the “results of the Aasen et al. [26] and Zhao et al. [7]”.

Zhao F, Wu H, Zhu S, Zeng H, Zhao Z, Yang X, Zhang S. Material stock analysis of urban road from nighttime light data based on a bottom-up approach. Environ Res, 2023; 228, 115902. https://doi.org/10.1016/j.envres.2023.115902

17.L340-341: “model will lead to an increase in the complexity of the model.” Add following reference here as “model will lead to an increase in the complexity of the model (Zheng et al., 2023; Yin et al., 2023). Add references according to the journal like [20, 21].

Zheng H, Fan X, Bo W, Yang X, Tjahjadi T, Jin S. A Multiscale Point-Supervised Network for Counting Maize Tassels in the Wild. Plant Phenomics. 2023; 5, 100. https://doi.org/10.34133/plantphenomics.0100

Yin L, Wang L, Li T, Lu S, Tian J, Yin Z., et al. U-Net-LSTM: Time Series-Enhanced Lake Boundary Prediction Model. Land, 2023; 12, 1859. https://doi.org/10.3390/land12101859

-> Answer: Thank you very much for your remind! The reviewer gave professional references and current up-to-date references, which were revised in the article, and the references provided improved my new understanding in this research area, thank you very much for the patient review!

18.Don’t repeat results in discussion section.

-> Answer: Thank you very much for your remind! We have removed some of the results in the discussion section!

19.In conclusion you just make repetition of similar information as given in the start of conclusion. Make it clear.

-> Answer: Thank you very much for your comments! We have simplified the conclusion section to describe it more clearly, thank you very much for your professional guidance!

In this study, a method utilizing unmanned aerial vehicle (UAV) remote sensing imagery is developed for estimating the SPAD (Soil Plant Analysis Development) of sugar beet canopies. The approach integrates field-measured data and aerial UAV remote sensing data, incorporating the selection of vegetation indices and the assessment of the correlation between SPAD and vegetation indices. Five machine learning algorithms, namely KNN (K-Nearest Neighbors), Lasso, Random Forest, Ridgecv, and SVM (Support Vector Machine), are employed for model inversion. Under normal irrigation conditions, the correlation coefficients (R2) between predicted and test values are 0.635, RMSE (Root Mean Square Error) is 2.13, and RE (Relative Error) is 0.80%. Under drought stress, the corresponding values are R2=0.609, RMSE=2.71, and RE=0.10%. The results of this study demonstrate that constructing prediction models using the SVM algorithm under different water treatments can effectively invert the chlorophyll content of sugar beet canopies with varying growth conditions.

20.Need to focus on the findings according to result. What are recommendations?

-> Answer: Thank you very much for your comments! We have scrutinized the full text and revised some of the descriptions where they were unclear.

21.Double check that all references are cited within the text

-> Answer: Thank you very much for your comments! We double-checked all the references and made changes according to the format of the journal.

Reviewer #2:

1.In lines 37-39 taxonomical details for the studied plant species are erroneous. Sugar beet (scientific name: Beta vulgaris) belongs to Amaranthaceae family and genus Beta and not as it is cited [genus Sugar Beet, family Quinoa]. The manuscript do not cite the scientific name of the studied plant species.

-> Answer: Thank you very much for your comments! Thank you very much for your professional opinion, we have also learned a lot by reviewing the literature and giving references in the article.

2.In lines 92-94 it is cited that [300 copies of domestic and foreign staple varieties were selected as the participating materials…]. There is no available information a) for their genetic and/or physiological background b) for their resistance to stress factor (drought). Further explanations are needed due to the fact that outcome UAV spectral data are not linked with known informative basis of the crop.

In Result section lines: 226-229 “In conclusion, the overall data dispersion and the range of variability are still large, and it also indicates that the population is rich in genetic variation. Genetic variation should be standardized, defined and stated prior to the experiment; authors received SPAD response variation in stressed and non-stressed conditions which cannot provide reverse statements for optimized materials experimentation.

3.Drone platforms and flight setting are well described; however no geographical spatial / positioning information was provided for the scanned area.

-> Answer: Thank you very much for your comments! Regarding the 300 populations you mentioned, 300 natural populations were used in this study for the SPAD model prediction study of sugar beets, and since the genotypes of the 300 populations were different, we analyzed the variability of the populations in the data analysis of the phenotypes. Thank you very much for your professional question. The next step of our plan is to correlate the genotypes of SPAD content in sugar beets with the phenotypes obtained by drone remote sensing to find out the genes that may affect it and to provide a theoretical basis for the subsequent breeding of sugar beet.

4.Authors provide extensive discussion on causes of chlorophyll decline due to drought (section 4.1-Discussion); however, this information is not taken into account on experimental planning and outcome results. Core aim of the manuscript is to introduce UAV remote sensing crop phenotyping in machine learning (as it is indicated in lines 79-87) but there is no categorization and or validation of drought-chrolophyll decline causes prior data being introduced to algorithmic – machine learning analysis phase.

-> Answer: Thank you very much for your professional evaluation, in the article we set up different treatments of water and drought to study the SPAD of sugar beets, as there are fewer studies on UAV SPAD of sugar beets at the moment, not much can be referred to in this part, we compared the prediction accuracy of the UAV model through water and drought and evaluated it, and the actual measurements were given as well, this is in the part of the analysis of the results.

5.Some inconsistency typing problems found citation’s section format (capital letters text etc).

-> Answer: Thank you very much for your comments! We double-checked all the references and made changes according to the format of the journal.

---

## [Editor Report · Decision Letter 1]

21 Feb 2024

Remote sensing estimation of sugar beet SPAD based on unmanned aerial vehicle multispectral imagery

PONE-D-23-33769R1

Dear Dr. Li,

We’re pleased to inform you that your manuscript has been judged scientifically suitable for publication and will be formally accepted for publication once it meets all outstanding technical requirements.

Kind regards,

Habib Ali, PhD

Academic Editor

PLOS ONE
---

## [Editor Report · Acceptance letter]

15 May 2024

PONE-D-23-33769R1 

PLOS ONE

Dear Dr. Li, 

I'm pleased to inform you that your manuscript has been deemed suitable for publication in PLOS ONE. Congratulations! Your manuscript is now being handed over to our production team.

Kind regards, 

on behalf of

Professor Habib Ali 

Academic Editor

PLOS ONE